# Youth Mental Health Peer Support Work: A Qualitative Study Exploring the Impacts and Challenges of Operating in a Peer Support Role

**Calvert Tisdale [1,\*], Nicole Snowdon [2], Julaine Allan [3]****, Leanne Hides [1], Philip Williams [4] and Dominique de Andrade [1]**

1   Lives Lived Well Research Group, National Centre for Substance Use Research, School of Psychology, The University of Queensland, Brisbane 4072, Australia; l.hides@uq.edu.au (L.H.); d.deandrade@deakin.edu.au (D.d.A.)
2   National Drug and Alcohol Research Centre, University of New South Wales, Sydney 1466, Australia; n.snowdon@unsw.edu.au
3   School of Health and Society, University of Wollongong, Wollongong 2500, Australia; julaine@uow.edu.au
4   Child and Youth Mental Health Services, Mental Health and Specialist Services, Gold Coast Hospital and Health Service, Gold Coast 4215, Australia; Philip.Williams@health.qld.gov.au
\*   Correspondence: c.tisdale@uq.edu.au; Tel.: +61-435-256-618

**Abstract:** Youth aged 16–24 years have the highest prevalence of mental illness in Australia, accounting for 26% of all mental illness. Youth mental health peer support work is a promising avenue of support for this population. However, limited research has examined impacts on those who provide youth mental health peer support work. We aimed to identify the benefits and challenges of working in a youth mental health peer support role. Semi-structured qualitative interviews with seven purposefully sampled peer workers from a national youth mental health organisation in Australia were conducted. The interviews were thematically analysed. Six key themes were identified: (1) personal growth, (2) interpersonal factors, (3) organisational factors, (4) boundaries, (5) role acknowledgement, and (6) challenging situations. Key supportive factors included financial reimbursement, training, support, and role-related flexibility. Identified challenges included lack of role acknowledgement, role-related stress, and boundaries. Operating within a youth mental health peer support role is perceived to have positive impacts on personal growth and interpersonal factors, enhanced through financial reimbursement, supervision, and role-related flexibility. Perspectives on the most effective form of role boundaries were diverse however their importance in addressing challenges was emphasised.

**Keywords:** peer support; peer work; peer support work; youth mental health; qualitative; consumer provider; youth

## 1. Introduction

According to the Australian Bureau of Statistics (2009; 2015) nearly half (45%) of all Australians will experience a mental illness in their lifetime, with around approximately one in nine (11.7%) people experiencing high or very high levels of psychological distress. Within this population, youth aged 16–24 years have the highest prevalence of mental illness, accounting for 26% of all mental illness [1]. The high prevalence of mental illness in Australia, particularly among young people suggests potential benefits of providing early intervention services for this vulnerable population. One avenue of mental health support for young people is peer support.

Peer support is social-emotional support with the goal of instigating social or personal change, delivered by individuals with a lived experience comparable to the persons they are aiding [2]. Present in mental health settings since the 1970's peer support roles have extended into traditional health settings [3,4]. Beginning as a consumer-led movement

critiquing traditional mental health practices, recovery-oriented practice shifts control from the professional to the service-user with the goal of improving well-being rather than meeting clinical outcomes [5]. Peer support work promotes person-centred recovery through the lived experiences of service users [6,7]. Adult peer support or consumer led services have proliferated because of recognition of the positive impact peer services have for service users and the health system [8,9]. With the increased presence of peer support workers in youth health care settings, there is a need to examine the impact of this model on both peer workers and those they support in youth health services.

The positive impacts of receiving peer support have been [4,10], however numerous systematic reviews point to a lack of high quality evidence and a need for more rigorous methodology [9,11–14]. Research about impacts on those who deliver peer support is limited. Recent meta-syntheses in the field have identified the peer role to positively impact on the peers' sense of purpose, self-acceptance, and employment opportunities, as well as their interpersonal skills and social networks [15,16] Furthermore, delivering peer support has been suggested to have a positive impact on perceived personal growth and recovery [10]), skill development [4], and relapse rates [15]. While the benefits of peer support work have been demonstrated in adult populations, further research is needed for young people in this role.

Identifying the barriers to delivering peer support effectively is equally as important. Systematic reviews and syntheses investigating the impact on peer workers operating in mental health services identified challenges to delivering peer support. Key challenges included personal health limitations, boundaries, and lack of training, supervision, and, financial compensation [17,18]. Effective implementation of peer support workers into organisations is key in addressing these barriers. In a systematic review on the influences on the implementation of peer support work for adult mental health problems, factors identified to enhance peer support worker implementation included role definition, training, wellbeing, and organisational culture [19]. Further investigation is needed to understand the impact these barriers may have on those operating specifically in youth peer worker mental health roles, and how these barriers can be addressed.

This study aims to address the identified limitations and gaps in existing literature on youth mental health peer workers, by qualitatively exploring the experiences of young people operating in youth mental health peer support roles. The study had two specific aims: (1) to explore the impact of the peer role on the mental health of those who provide youth mental health peer support; and (2) to identify the benefits and challenges experienced by those who provide youth mental health peer support.

## 2. Materials and Methods

Headspace is an Australian Government funded national youth mental health service for ages 12 to 25 years. Mental health services provided at Headspace include health professionals, information services, and peer support programs. At the time of data collection, headspace Southport in Queensland had run a peer support model for approximately two and a half years as part of the headspace Early Psychosis program. Headspace Southport utilised a non-contractual volunteer reimbursement model wherein peers volunteered their time and were financially reimbursed on an hourly rate. A role description, training program, and support structure was developed for the role by the specialist youth engagement coordinator at the centre. Peer workers attended clinician managed groups, workshops, and supervised activities (e.g., physical or educational) to support and represent young people who attend the service. This role involved sharing aspects of their lived experience, engaging with young people one-on-one, and providing targeted support.

A qualitative approach was chosen to explore the peer work experience. Semi-structured interviews were used to explore participants' experience of the peer role within the youth mental health service. Interviews were professionally transcribed and analysed using the thematic analysis procedures described by Braun & Clarke [20]. With a

semantic focus, a realist thematic analysis perspective was taken, assuming a unidirectional relationship of language between meaning and experience.

Recruitment was limited to headspace Southport due to the unique structure of their youth mental health peer support worker program. Potential participants were either a current or former youth mental health peer support worker at the centre. Sixteen current (*n* = 6) and former (*n* = 10) peer workers were informed about the study and invited to contact the researcher. Seven of the sixteen participated in the study. Of the nine that did not participate, three were overseas or interstate, two were acutely unwell, two were uncontactable, one was unable to find time, and one did not respond to the invitation. Peer participants' demographic characteristics are presented in Table 1.

**Table 1.** Demographic characteristics of participants.

| Variable | N | % |
|:---:|:---:|:---:|
| Age (*M* = 23.57, *SD* = 3.46) | | |
| ≤ 24 years | 6 | 86% |
| >24 years | 1 | 14% |
| Gender | | |
| Male | 1 | 14% |
| Female | 6 | 86% |
| Highest level of education | | |
| High school | 1 | 14% |
| TAFE/College Certificate | 1 | 14% |
| University Degree | 5 | 72% |
| Time in peer role | | |
| Less than 6 months | 1 | 14% |
| 6 months–1 year | 1 | 14% |
| 1 year–less than 2 years | 4 | 58% |
| 2 years | 1 | 14% |
| Peer role status | | |
| Current Peer | 5 | 71% |
| Past Peer | 2 | 29% |
| Diagnoses (not-exclusive) | | |
| Affective Disorder | 4 | 58% |
| Anxiety Disorder | 4 | 58% |
| Eating Disorder | 1 | 14% |
| Obsessive Compulsive Disorder | 1 | 14% |
| Post-Traumatic Stress Disorder | 2 | 29% |
| Psychotic Disorder | 1 | 14% |
| Substance Use Disorder | 1 | 14% |

The study was approved by The University of Queensland, School of Psychology Ethical Review Committee, clearance number: 18-PSYCH-4-119-JMC. Written consent was obtained from participants. Interviews lasted between 30 and 70 min with most interviews running for an hour. All interviews were conducted in private consultation rooms at headspace Southport, during June and July 2018. Participants were financially reimbursed for their time.

All interviews were audio recorded and de-identified before being transcribed verbatim by a professional transcriptionist, which were checked against the recording for accuracy by the first author (CT). Interview questions (Appendix A) explored the benefits and challenges of the peer role including impacts on the participant's mental health, relationships with service users and clinicians, and organisational factors related to the role.

Qualitative data were analysed using NVIVO 11, adhering to Braun and Clarke's [20] guide to thematic analysis (see Appendix B). An initial codebook was developed through coding of one transcript by two authors (CT and NS). This initial codebook was used to iteratively develop a final codebook through constant comparison and discussion of two further transcripts. The final codebook was used by CT to code all transcriptions and

identify themes. NS used this final codebook to code two further interviews to reduce researcher bias and introduce inter-reliability.

In accordance with Braun & Clarke's (2006) steps, the dataset was examined in its entirety to provide a rich thematic description. This methodological approach is useful for an under-researched area despite sacrificing depth and complexity. Themes were focused semantically by identifying the explicit meanings of the data. Interpretations were summarized to theorise the significance of the broader meanings and implications of patterns in the data. Epistemology was approached from a realist thematic analysis perspective, motivations and experiences were theorised in a straight-forward manner as a unidirectional relationship of language between meaning and experience was assumed.

CT conducted the interviews and was the primary analyst and coder. CT was independent of headspace in all capacities at time of data collection. NS was the secondary coder, independent to the data collection phase and independent of headspace in all capacities.

## 3. Results

Seventeen codes were derived from the dataset with six overarching themes emerging from these codes (refer to Table 2 for full list of codes and themes).

**Table 2.** List of identified codes and overarching themes.

| Theme One: Personal Growth | Theme Two: Interpersonal Factors | Theme Three: Organisational Factors | Theme Four: Boundaries | Theme Five: Role Acknowledgement | Theme Six: Challenging Situations |
|---|---|---|---|---|---|
| 1. Validation, normalisation, and self-acceptance<br>2. Purpose, direction, and perspective<br>3. Positive impacts and growth<br>4. Professional growth | 1. Social networks<br>2. Unique role and relationship<br>3. Young People | 1. Flexibility<br>2. Financial reimbursement<br>3. Support, debriefing and training | 1. Uncertainty<br>2. Prevent helping/Inability to help<br>3. Equality of relationship | 1. Viewed as patient and not staff<br>2. Lack of staff acknowledgement<br>3. Role ambiguity | 1. Challenging or stressful situations |

Three of the key themes related to the perceived impact of the peer role on peer support workers: *Personal Growth, Interpersonal Factors, and Organisational Factors*. The three remaining key themes pertained to factors that were perceived to either present as barriers or challenges to the peer role and peer workers themselves: *Boundaries, Role Acknowledgement, and Challenging Situations*.

### 3.1. Personal Growth

Peer workers felt that the role fostered personal growth and development. Participants believed that this pertained to not only their beliefs and feelings of personal growth but also development related to skills and opportunity:

> *I feel like the role has given me way more benefits than it has given me [negatives] because it's given me more confidence, more ability to share my story comfortably, being able to be vulnerable around people without the fear of them judging me.* (#7)

Several of the peer workers had completed undergraduate studies in Psychology and saw the peer work as an opportunity to develop skills in this area and develop professionally:

> *I think it maybe gave me a little bit more of purpose with mental health roles and understanding myself and where I wanted to work, like more career wise.* (#5)

Interactions with both young people and other peers were facilitators to self-acceptance, validation, and normalisation of their own lived experiences.

> *Because for me being able to share my experiences and open up to other young people and hear them come back and say hey I've had that experience too. It's really validating.* (#2)

A small number of peer workers felt that being in the role provided perspective to compare their current and past self, reflect on their lived experience, and gain new insights on their own recovery.

*I think in some ways it helped because it made me realise my progress, which sounds really cheesy but that's true.* (#7)

Similarly, all peer workers discussed the rewarding nature of the role and its contribution to a feeling of purpose, with some peers believing this had positive implications for their sense of value and mental health.

*So it's all positives, it was all positives coming at me because of what I'm putting out is my 100 percent. So it made me feel even better about myself [ . . . ]* (#1)

Peers viewed the personal growth gained from the peer role to be a positive impact on their wellbeing and mental health when they perceived they were completing their work with young people successfully.

*3.2. Interpersonal Factors*

The peer workers highlighted the importance of their relationships with young people, non-peer staff and other peer workers. They described how the role had expanded their social networks and facilitated the development of unique relationships with young people. The importance of the social network of peer workers that were able to understand, support and relate to each other both within and outside the peer role was also highlighted:

*I guess because we're in the same role, so if something goes on for one of them we can all listen and be understanding but also because we're a similar age, if something else goes on outside of headspace with our life then we can tell the other person and still get support.* (#5)

All peers described headspace as a positive, safe, and supportive environment they enjoyed working in, often due to their positive interactions with non-peer staff:

*[ . . . ] we are I guess another staff member here and we are seen in that way by the staff. Our unique role is valued by them and we have got a desk out the back and we can use the staff facilities.* (#3)

Moreover, the peer workers felt that working in a positive environment removed pressure related to their role and allowed them to provide peer support more effectively:

*So I really do enjoy doing this work and I really enjoy coming to work especially at headspace because it's really such a positive environment anyway. Everything's just so positive [ . . . ] everything here does feel like a big family and it's really nice.* (#4)

Peer workers also identified their unique relationship with young people as important. All peers believed having a less clinical and more equal relationship with young people was an advantage that often resulted in greater disclosure:

*I guess that I can kind of—it can be a bit more of an equal relationship where we can both share information with each other. Yeah, I guess like I'm not sitting down writing notes about them afterwards.* (#3)

This unique relationship that floated between a professional and non-professional relationship allowed peers to relate, guide, and support young people based on their lived experience. Peers felt this positioned them uniquely within the organisation, contributing to their feelings of legitimacy and value.

*3.3. Organisational Factors*

Peer workers discussed organisational factors that impacted on their role. Peers felt that the flexibility of the role afforded by the volunteer reimbursement model allowed them to prioritise their own recovery and personal lives when necessary. Autonomy surrounding their work allowed for open communication between peer workers and the organisation.

All peers discussed the flexibility of the role and feeling empowered to request assistance from the organisation or other peers:

> *I think for mental health reasons it's important to be able to just take time for yourself and to make sure that you're taking care of other parts of your life.* (#5)

The volunteer reimbursement model afforded flexibility within the peer role, prompting collaborative teamwork among peer workers:

> *[ . . . ] I personally feel valued by the team in the workplace and so then in return I want to make myself and available and flexible and value whatever they're doing as well and how my role might fit into it.* (#3)

While not viewed as integral, most peers perceived financial reimbursement as an important factor that contributed to acknowledgement of the peer role, contributing to a sense of legitimacy, credibility and value within the organisation:

> *I think it made the role, not like official, but I guess us feel like, or it made me feel like I was somewhat worthy or important.* (#6)

Having a dedicated supervisor in the youth engagement coordinator position who was understanding and trusting enabled peers in the role to feel valued and supported. Participant one stated:

> *She [supervisor] said before to all the peer workers, if there's anything you come up against that's related to your mental state, if you're coping or not, come see me and we can talk about it, or someone will be able to talk about it with you.*

Similarly, by being trained and conducting role-plays of potential situations, peers felt more confident and prepared in their role, leading to reduced role ambiguity.

> *They did do a really good induction where we went through a couple of scenarios that could have come up.* (#6)

Peers felt debriefing and training provided by the organisation facilitated opportunities to develop their skills and discuss any issues:

> *[ . . . ] any things that we found uncomfortable, any improvements, anything we'd like to add, and we'd kind of go round and share each of our stuff, either an issue that's passed or one that we're dealing with and we're talking about how we can resolve it.* (#7)

Peers viewed the model employed by the service provider as critical in how they conducted their work and were impacted by their role. Training, supervision, and financial reimbursement were identified as key organisational factors that enhanced the peer role.

### 3.4. Boundaries

All peers believed the role necessitated sharing of personal experiences and viewed maintaining interpersonal boundaries as an intrinsic element of the role. The nature of boundaries was viewed diversely among peer workers. Some defined their peer work as a unique role that was positioned between friendship and a non-peer staff relationship, where sharing their lived experiences was fundamental.

> *My job is just to I guess provide support and help them feel like they're not alone in what they are going through [ . . . ] I would definitely say the relationship is definitely more a friendship than a professional relationship.* (#4)

Other peers discussed the importance of being able to positively assert both professional and personal boundaries in their role, highlighting the importance of autonomy and security when sharing their lived experience:

> *I think that's kind of harder to swing if you're employed as well because it's like this is part of your role you should have to do this. Whereas with us it's kind of like it's up to you, you can choose a bit more.* (#5)

Peer workers often reported that boundaries were an effective tool in approaching young people who misconstrued the relationship as a friendship to preserve the peer-to-peer relationship. Participant three gave a specific example of situations where asserting boundaries aided in navigating difficult situations that arose from the role.

> *[ . . . ] they relate to me enough to want to hang out with me outside of work, but then you have to say "I'd love to but unfortunately because I am in this role I can't hang out with you outside of headspace. (#3)*

Some peers discussed challenges operating within the peer work service model when they were unable to offer more to clients due to role boundaries preventing them from providing further support:

> *Knowing that at the end of that eight weeks, I couldn't do anything else. That was it, that's such, and that made me feel pretty shitty. (#2)*

These peers felt that while this inability to assist was unhelpful, it was necessary to have boundaries in place to avoid over-involvement or burnout. Peers also discussed feelings of frustration when young people presented resistance to the boundaries of the peer relationship:

> *So it's really challenging watching a young person not have that attitude about it and not wanting to get help or not wanting to change or not having that positive attitude. (#4)*

Peers discussed the importance of feeling supported in their role boundaries and believed this was achieved through supervision:

> *It's really good because we debrief, so if I am feeling really shit [ . . . ] Then they sit down and talk to me about it. We had an experience with a young person [ . . . ] We debriefed about that and then there was so much follow up afterwards to make sure that I was okay, so just knowing that those support networks are there. (#2)*

Despite the multiple approaches to boundaries observed among peers, navigating these boundaries with young people was a common challenge. All peers believed supervision to be a key factor in addressing and managing these concerns.

### 3.5. Role Acknowledgement

Working within a service that provides professional services, peer workers were required to manage relations with non-peer staff such as clinicians. Most peers alluded to feelings that non-peer staff did not initially acknowledge them as equals or were unaware of their role in the organisation:

> *It felt a bit awkward and you didn't really belong in the staff area sometimes. (#6)*

However, participant seven viewed this as a positive aspect, as it allowed the unique relationship between peers and young people to feel less professional and more informal:

> *But then in terms of the employees, I feel like—like they really value us but they don't see us as equals. But I don't think that's a bad thing; I think that's just—they've just got a different role. That's how I see it. So I'm not bothered by it. (#7)*

Some peers expressed that any uncertainty was due to non-peer staff not understanding what the peer role entailed. Participants felt that some non-peer staff lacked confidence in the peer staff for example, participant six believed:

> *They [thought] we wouldn't be very beneficial for them [service users] or that we might give them the wrong advice. Or come off as being too chummy with them, rather than setting goals with them. (#6)*

Despite this, participant six discussed that time and exposure were important factors in non-peer staff coming to see the unique value the peer work force contributed, improving feelings of equality.

*I definitely feel like it has improved [ . . . ] maybe realising that [we] do help run the groups. But I think that might have just been something that happens over time. But yeah, and a bit of knowledge around what we were actually doing.* (#6)

Peers felt that the advocacy of a supervisor bridged the gap between non-peer staff and helped inform non-peer staff about the importance of the peer role. Participant six described this:

*[Supervisor] was really, really good with supervision. Finding out how the other staff treated us during groups as well [ . . . ] she was really good at giving us an equal footing [ . . . ]* (#6)

While a lack of role acknowledgement was recognised by peers, financial reimbursement and supervision were factors that approached this challenge through improving feelings of value and equality in peers.

*3.6. Challenging Situations*

A main challenge that was discussed by some peer workers was being placed in challenging or stressful situations during the role. Situations that arose from young people could often be present as challenges for peers. Participant gave an example of a situation:

*[ . . . ] we watched this movie and it was really full on where this guy was having a psychotic episode [...] it hit a nerve with me and I was like, oh I don't know if I can sit through any more of that. I just did and just went through it.* (#1)

These situations presented as opportunities for peers and the service provider to provide supervision and debriefing to work towards improving practices through collaboration and reflection. For example, participant one explained the role of supervision in ensuring their incident was addressed, and participant three describes the process following another separate challenging situation:

*Then I saw [supervision] straight away, she just approached me to tell me something [ . . . ] and I'm like, oh, I'm a bit spun out. She was like, okay no, we're dealing with it now [ . . . ]* (#1)

*Then we, in that session, brainstormed what our responsibilities are, what the facilitator's responsibility should be, what we expect from our end help wise and with boundaries and things. That was then passed on higher up the chain and things in groups have been better now.* (#3)

Peers also believed the volunteer reimbursement model wherein peers were able to nominate the hours they operated and what tasks they undertook was an important factor in avoiding challenging situations:

*So there's no pressure to do it if you don't feel like it because you know there's like three other people that will step in and be really happy to do it.* (#5)

While challenging situations commonly arose due to the nature of the peer support role, they provided opportunities for collaboration between peers and the service provider to debrief, reflect, and enhance practices. Flexibility and teamwork within the role were factors that were important in addressing these situations, facilitated by a volunteer reimbursement model.

## 4. Discussion

This study explored the impacts of operating in a peer support worker role in a youth mental health service. Identified in the core themes, peer workers felt the role encouraged personal growth while also presenting opportunities to develop their interpersonal skills. Organisational factors such as reimbursement, professional support, and opportunities for critical reflection on boundaries were important to sustaining the peer role.

Consistent with other studies, the peer role was perceived to provide both personal and professional development through facilitating self-acceptance, perspective and nor-

malisation while also providing opportunities to develop professionally relevant skills. Enhancing feelings of acceptance and purpose, and improvements in employment opportunity are opportunities created through the peer role [15]. Peer workers identified the importance of drawing on their own mental health experiences to help others, which has been identified as one of the key benefits of delivering peer support [10]. Operating within a peer role can development interpersonal skills and social networks [16]. Young people with lived experience of mental illness frequently experience stigma and exclusion from employment [21]. This study found that peer support is a way to validate peer experience, use it to support others and value the way it can create employment opportunities.

Some mental health impacts of peer support work were identified. Previous research has suggested that peer workers are vulnerable to stressors, role ambiguity, and burnout [17,18]. Consistent with findings on optimising peer worker implementation [19], peer workers in the current study felt factors such as organisational role acknowledgement, supervision, and training, enhanced integration, and effectiveness within the organisation. Peers also identified that financial reimbursement and flexibility provided opportunities to reduce challenges faced in their work in that they were viewed as part of the organisation's service delivery.

Challenging situations encountered in the role had the potential to lead to negative outcomes. While the key strength of peers is their lived experience, this can bring social, emotional, and physical limitations [17]. Risk of burnout or social and emotional harm caused by the peer role is a serious potential consequence both peers and organisations should addressTo facilitate and enhance the role of peer workers a need for collaboration and reflection among peer staff and non-peer staff was highlighted. A lack of role acknowledgement from non-peer staff was a barrier to operating in their role. Pressure of acceptance from the organization occurred when peers felt undervalued or lacking in credibility. Dedicated supervision assisted with navigating these relations, leading to increased feelings of value and credibility in the organisation. Though the form of boundaries was perceived diversely, their importance was emphasised and identified as a strategy for addressing challenging situations and client relationships. This aligns with prior research highlighting the importance of respecting and adhering to boundaries to reduce risk of burnout [18].

### 4.1. Strengths and Limitations

The qualitative methodology allowed exploration and analysis of the participant's experiences without sacrificing context or complexity. By employing semi-structured individual interviews, the present study ensured that information-rich data was collected, which has the potential to be more compelling and explanatory than quantitative data. While the peer worker coordinator was consulted and involved in the development of interview questions; future research should consist of greater involvement of peer staff.

A key limitation is the small sample size. Due to the nature of youth mental health peer support work, more than half of the individuals who engaged in this role were unavailable for interviewing. Despite the small number of participants, the current study approached all possible participants of the purposive sample strategy. Positive bias towards the headspace Southport peer worker program may have been present in the interviews as all but one participant had ongoing relationships with the headspace centre. Those who were unable to participate in interviews were all former peer workers and may have held views about the peer role that may not be consistent with peers in this study. Thus, the current study only exemplifies the perspectives of current but not former peers.

### 4.2. Implications for Future Research

This qualitative study conducted an examination of youth mental health peer workers, an under-represented population in an under-researched area. Results identified the potential personal growth and interpersonal benefits of peer support work, and the importance of strict peer-to-peer boundaries, dedicated supervision, financial reimbursement, and role-

related flexibility for supporting the work of peer workers. Barriers to operating effectively in the peer role included lack of role acknowledgement and risks of harm from burnout, highlighting the importance of advocacy and supervision. High quality mixed-method research with a greater sample size is required to further increase understanding of the benefits and challenges of peer support programs for both the providers and recipients, as well as the services involved. Nevertheless, the results of this study provide an important first step for comprehending the impacts, both positive and negative, of operating in a youth mental health peer role.

**Author Contributions:** Conceptualisation, C.T., D.d.A. and L.H.; methodology, C.T. and D.d.A.; formal analysis, C.T., N.S. and J.A.; investigation, C.T. and D.d.A.; resources L.H. and P.W.; data curation, C.T.; writing—original draft preparation, C.T. and D.d.A.; writing—review and editing, C.T., N.S., J.A., L.H., P.W. and D.d.A.; supervision D.d.A. and L.H.; project administration C.T. and D.d.A., funding acquisition P.W., L.H. and D.d.A. All authors have read and agreed to the published version of the manuscript.

**Funding:** Funding from Gold Coast Primary Health Network to complete professional transcriptions.

**Institutional Review Board Statement:** The study was conducted according to the guidelines of the Australian National Statement on Ethical Conduct in Human Research approved by The University of Queensland, School of Psychology Ethical Review Committee, clearance number: 18-PSYCH-4-119-JMC.

**Informed Consent Statement:** Informed consent was obtained from all subjects involved in the study.

**Acknowledgments:** Headspace Southport, their team of youth mental health peer support workers and their peer worker coordinator. Funding from Gold Coast Primary Health Network to complete professional transcriptions.

**Conflicts of Interest:** Calvert Tisdale completed this project as his thesis as part of acquiring his Bachelor of Psychological Science (Hons) at the University of Queensland and is currently a research assistant with the Lives Lived Well Research Group. Dominique de Andrade was an industry funded (Lives Lived Well) Research Fellow at University of Queensland at the time of data collection. Leanne Hides holds a National Health and Medical Research (NHMRC) Council funded senior research fellowship and is the Lives Lived Well Chair in Alcohol, Drugs and Mental Health. Julaine Allan was the Research Manager for Lives Lived Well at the time of data collection. Nicole Snowden was Research Project Coordinator for Lives Lived Well at time of data collection. Philip Williams was the Manager of headspace Southport at the time of data collection.

## Appendix A. Interview Questions

DEMOGRAPHIC

1. What is your age?
2. What is your gender identity?
3. How long have you been a peer support worker at Headspace?
4. Could you talk about your education or any qualifications you have?

LIVED EXPERIENCE/MENTAL HEALTH

1. A part of your role as a peer support worker is your experience with mental illness and being able to share your experiences with others. Would you feel comfortable talking about your lived experience with me?
2. How are things for you currently in regards to your mental health?
3. Do you think undertaking peer support work has had any impact on your mental health?
   a. Anything that has benefited your mental health?
   b. Anything that has presented a challenge to your mental health?
      i. How you overcame the challenges?

HEADSPACE

1. Could you please rate the following peer support work factors on their importance on a scale from 1 (not important) to 10 (highly important)
   a. Financial Reimbursement
      i. Getting reimbursed for your peer support work
   b. Flexibility
      i. Choosing what duties you undertake as a peer support worker
      ii. Choosing the hours that you work
      iii. Being able to assert your boundaries in terms of giving support (type of support and how much you share)
      iv. Taking time out for other commitments you may have
      v. Taking time out for your own health
2. You said that you found _____ important. Could you please explain why?
3. You said that you found _____ not that important to you. Could you please explain why?
4. Would you still undertake your role as a peer support worker if the things you found important were no longer available to you?
   a. Do you think your mental health would be impacted if you didn't have these things?
5. How valued do you feel in your role on a scale of 1 (not valued) to 10 (highly valued)?
6. How supported do you feel on a scale of 1 (not supported) to 10 (highly supported)?
   a. You said that you feel _____ in regards to feeling valued/supported. Could you please explain why?
7. How do you think your relationship with young people is different to other employed staff?

   PREVIOUS HEADSPACE WORKERS

1. How long were you a peer support worker?
2. What are you doing now?
3. Has the role informed what you do now? How so?

### Appendix B. Processes of Braun & Clarke's (2016) Thematic Analysis

| Phase | Description of the Process |
| --- | --- |
| 1. Familiarising yourself with your data: | Transcribing data (if necessary), reading and re-reading the data, noting down initial ideas. |
| 2. Generating initial codes: | Coding interesting features of the data in a systematic fashion across the entire data set, collating data relevant to each code. |
| 3. Searching for themes: | Collating codes into potential themes, gathering all data relevant to each potential theme. |
| 4. Reviewing themes: | Checking if the themes work in relation to the coded extracts (Level 1) and the entire data set (Level 2), generating a thematic 'map' of the analysis. |
| 5. Defining and naming themes: | Ongoing analysis to refine the specifics of each theme, and the overall story the analysis tells, generating clear definitions and names for each theme. |
| 6. Producing the report: | The final opportunity for analysis. Selection of vivid, compelling extract examples, final analysis of selected extracts, relating back of the analysis to the research question and literature, producing a scholarly report of the analysis. |

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
