# Peer review of "Youth Mental Health Peer Support Work: A Qualitative Study Exploring the Impacts and Challenges of Operating in a Peer Support Role"

_adolescents, doi:10.3390/adolescents1040030_

Round 1
Reviewer 1 Report
Thank you for this interesting qualitative study that can be regarded as a stepping stone for further research. While a lot of the limitations are addressed in the study, I do think that some elements need further clarification:
- The fact that this is done with people related to just one organization is an important limitation. Maybe it's interesting therefore to discuss the working of this organization more in-depth.
- It's true that the amount of participants is small, but also relevant is the question of saturation was achieved or not?
- Make more explicit how you've worked on reliability and validity (You already mentioned a lot but you should elaborate more.
Looking at the findings there is a certain mismatch between what is promised at the beginning of the study:
-
Identifying the barriers to delivering peer support effectively is equally as important
But this is less present in the discussion and the conclusion that talk more about the benefits and possibilities of peer support, as you summarize like this:
-
Results identified the potential personal growth and interpersonal benefits of peer support work, and the importance of strict peer-to-peer boundaries, dedicated supervision, financial reimbursement, and role-related flexibility for supporting the work of peer workers.
Answering this original extra question in the discussion/conclusion could make the paper more relevant as I do think that you have elements to discuss this more profoundly, e.g. being well of not accepted in the role of peer supporter by professionals.
Author Response
Reviewer 1
Thank you for this interesting qualitative study that can be regarded as a stepping stone for further research. While a lot of the limitations are addressed in the study, I do think that some elements need further clarification:
- The fact that this is done with people related to just one organization is an important limitation. Maybe it's interesting therefore to discuss the working of this organization more in-depth.
[Response]: We thank the reviewer for reiterating the importance of the proposed article. We acknowledge and agree that the study focusing on a singular organisation presents an important limitation. We wished to investigate this unique youth mental health peer support service which remains novel to the present day and investigate how this research could be used as a steppingstone to understand a novel method of youth mental health support services. We agree the organisation plays an important role and have enhanced our description of the organisation and youth mental health peer support program in the Materials and Methods section.
“Headspace is an Australian Government funded national youth mental health service for ages 12 to 25 years. Mental health services provided at Headspace include health professionals, information services, and peer support programs. At the time of data collection, headspace Southport in Queensland had run a peer support model for approximately two and a half years as part of the headspace Early Psychosis program. Headspace Southport utilised a non-contractual volunteer reimbursement mod-el wherein peers volunteered their time and were financially reimbursed on an hourly rate. A role description, training program, and support structure was developed for the role by the specialist youth engagement coordinator at the centre. Peer workers attended clinician managed groups, workshops, and supervised activities (e.g. physical or educational) to support and represent young people who attend the service. This role involved sharing aspects of their lived experience, engaging with young people one-on-one, and providing targeted support.” [page 2, line 82-93]
- It's true that the amount of participants is small, but also relevant is the question of saturation was achieved or not?
[Response]: While saturation (continuing data analysis until no new information appears) is a commonly approached concept in qualitative research, this was not appropriate for the current study which interviewed all available members of the purposive sample. Therefore, saturation for this sample size is defensible as being sufficient in providing understanding, depth, and complexity of the group of interest.
- Make more explicit how you've worked on reliability and validity (You already mentioned a lot but you should elaborate more.
[Response]: To emphasise reliability, we have included:
“The final codebook was used by CT to code all transcriptions and identify themes. NS used this final codebook to code two further interviews to reduce researcher bias and introduce inter-reliability.” [page 4, line 164-165]
To explore validity, we have expanded upon our perspective and qualitative approach in accordance with Braun & Clarke (2006):
“In accordance with Braun & Clarke’s (2006) steps, the dataset was examined in its entirety to provide a rich thematic description. This methodological approach is useful for an under-researched area despite sacrificing depth and complexity. Themes were focused semantically by identifying the explicit meanings of the data. Interpretations were summarized to theorise the significance of the broader meanings and implications of patterns in the data. Epistemology was approached from a realist thematic analysis perspective, motivations and experiences were theorised in a straight-forward manner as a unidirectional relationship of language between meaning and experience was assumed.” [page 4, line 166-173]
Looking at the findings there is a certain mismatch between what is promised at the beginning of the study:
- Identifying the barriers to delivering peer support effectively is equally as important
But this is less present in the discussion and the conclusion that talk more about the benefits and possibilities of peer support, as you summarize like this:
- Results identified the potential personal growth and interpersonal benefits of peer support work, and the importance of strict peer-to-peer boundaries, dedicated supervision, financial reimbursement, and role-related flexibility for supporting the work of peer workers.
Answering this original extra question in the discussion/conclusion could make the paper more relevant as I do think that you have elements to discuss this more profoundly, e.g. being well of not accepted in the role of peer supporter by professionals.
[Response]: We agree that highlighting barriers within this unique peer role could improve the relevance of the paper. We have emphasised these results in the discussion and conclusion to highlight the barriers. We have included in the discussion:
“Challenging situations encountered in the role had the potential to lead to negative outcomes. While the key strength of peers is their lived experience, this can bring social, emotional, and physical limitations [17]” [page 10, line 472-474]
“A lack of role acknowledgement from non-peer staff was a barrier to operating in their role. Pressure of acceptance from the organization occurred when peers felt undervalued or lacking in credibility.” [page 10, line 477-480]
In implications for future research, we have highlighted these important barriers:
“Barriers to operating effectively in the peer role included lack of role acknowledgement and risks of harm from burnout, highlighting the importance of advocacy and supervision.” [page 11, line 507-509]
Reviewer 2 Report
The topic was interesting, but the very small sample size and predominantly subjective nature of the results limits applications in other settings. Thank you for raising this important issue of peer support in mental health work.
Author Response
Reviewer 2
The topic was interesting, but the very small sample size and predominantly subjective nature of the results limits applications in other settings. Thank you for raising this important issue of peer support in mental health work.
[Response]: We thank the reviewer for addressing the importance of the proposed paper. We agree the study has limited generalisability to other settings due to the subjectivity and small sample size. Generalisability is not always an expected attribute of qualitative research however qualitative research has been shown to generalise to other settings, even if only in a limited capacity. We believe that despite these limitations, this research is an important steppingstone into understanding this novel method of youth mental health peer support work and may lead to the promotion of such services in similar settings.
We have adjusted implications for future research to reflect this:
“High quality mixed-method research with a greater sample size is required to further increase understanding of the benefits and challenges of peer support programs for both the providers and recipients, as well as the services involved.” [page 11, line 509-512]
Reviewer 3 Report
This paper is quite original and presents compelling information that is of interest to readers. However, the findings are limited in significance by the very small sample size, and especially, the lack of a majority of the peer support workers being interviewed in this one location. as such, the contribution to scholarship is low and the academic soundness of the findings limited.
Having said that, the methods are quite original and with a larger sample size, the findings would be of much greater significance. This limitation is addressed in the paper, though, and I think the findings are of enough interest that publication is warranted.
I would encourage the authors to consider means of expanding the sample size and presenting a version of these results with far more peers.
Author Response
Reviewer 3
This paper is quite original and presents compelling information that is of interest to readers. However, the findings are limited in significance by the very small sample size, and especially, the lack of a majority of the peer support workers being interviewed in this one location. as such, the contribution to scholarship is low and the academic soundness of the findings limited.
Having said that, the methods are quite original and with a larger sample size, the findings would be of much greater significance. This limitation is addressed in the paper, though, and I think the findings are of enough interest that publication is warranted.
I would encourage the authors to consider means of expanding the sample size and presenting a version of these results with far more peers.
[Response]: We thank the reviewer for highlighting the originality and importance of the proposed paper. We agree the study has limited generalisability to other settings due to an inability to capture the majority of the purposive sample. Generalisability is not always an expected attribute of qualitative research however qualitative research has been shown to generalise to other settings, even if only in a limited capacity. While we agree that with a larger sample size the findings could be of greater significance, the nature of youth mental health peer support as a novel method limited these possibilities during the inception of the research. With greater understanding and adoption of this method future research could investigate with greater sample sizes.
We have adjusted implications for future research to reflect this:
“High quality mixed-method research with a greater sample size is required to further increase understanding of the benefits and challenges of peer support programs for both the providers and recipients, as well as the services involved.” [page 11, line 509-512]